# Edge states and skyrmion dynamics in nanostripes of frustrated magnets

A.O. Leonov[1] & M. Mostovoy[1]

Magnetic skyrmions are particle-like topological excitations recently discovered in chiral magnets. Their small size, topological protection and the ease with which they can be manipulated by electric currents generated much interest in using skyrmions for information storage and processing. Recently, it was suggested that skyrmions with additional degrees of freedom can exist in magnetically frustrated materials. Here, we show that dynamics of skyrmions and antiskyrmions in nanostripes of frustrated magnets is strongly affected by complex spin states formed at the stripe edges. These states create multiple edge channels which guide the skyrmion motion. Non-trivial topology of edge states gives rise to complex current-induced dynamics, such as emission of skyrmion–antiskyrmion pairs. The edge-state topology can be controlled with an electric current through the exchange of skyrmions and antiskyrmions between the edges of a magnetic nanostructure.

[1] Zernike Institute for Advanced Materials, University of Groningen, Nijenborgh 4, 9747 AG Groningen, The Netherlands. Correspondence and requests for materials should be addressed to A.O.L. (email: leonov@hiroshima-u.ac.jp).

Chiral magnets, that is, magnets with a non-centrosymmetric crystal lattice, show a variety of non-collinear magnetic states stabilized by the relativistic Dzyaloshinskii–Moriya (DM) interaction. The recent discovery of skyrmions in chiral magnets[1,2] led to many theoretical and experimental studies of unusual physical properties of these topological excitations[3]. Low critical currents required to set skyrmions into motion[4,5] opened a new active field of research in memory and logic devices, in which information is carried by skyrmions[6–11].

The DM interaction imprints the chirality of crystal lattice into the chirality of magnetic orders: the direction of spin rotation in spirals and skyrmions is determined by the lattice. Chiral magnetic states can also originate from competing ferromagnetic and antiferromagnetic exchange interactions between spins in Mott insulators. It was recently shown that frustrated magnets form a new class of materials that can host skyrmion crystals and isolated skyrmions[12,13]. In contrast to DM interactions, exchange interactions are insensitive to the direction of spin rotation in non-collinear magnetic states, which gives skyrmions two additional degrees-of-freedom—vorticity and helicity. In frustrated magnets, skyrmions coexist with antiskyrmions and carry a reversible electric dipole moment[13,14]. Magnetic frustration is not limited to Mott insulators: RKKY interactions[15], competing double exchange and superexchange interactions[16], double exchange in charge-transfer systems[17] and fluxes of effective magnetic fields[18–20] can stabilize non-collinear and even non-coplanar magnetic states in itinerant magnets.

Here, we explore the current-induced dynamics of skyrmions and antiskyrmions in nanostripes of frustrated magnets and show that it is strongly affected by the periodically modulated spin structures formed at the stripe edges. These edge states are topological and have a highly nonlinear dynamics of their own: under an applied electric current they emit and absorb skyrmions and antiskyrmions. These processes, governed by a topological conservation law, allow for the electric control of edge-state topology.

## Results

**Edge states and edge channels in frustrated magnets.** The nontrivial skyrmion topology gives rise to a high energy barrier that prevents the decay of skyrmions into magnons. However, near the boundaries of a magnet this barrier can be significantly lower or may not exist at all. Therefore, the practical use of skyrmions crucially depends on their repulsion from edges of magnetic nanostructures. In chiral magnets such a repulsion is naturally provided by the bulk DM interaction, which tilts the magnetization vector away from the magnetic field direction at the edges of a magnet, giving rise to the so-called edge states[21–24].

Competing spin interactions in frustrated magnets do not necessarily induce similar spin tilts. However, exchange interactions and magnetic anisotropies at surfaces or interfaces of magnetic materials can be significantly different from those in bulk, because of a lower symmetry of magnetic ions at the edges[25–27]. We have found that a strong surface anisotropy gives rise to edge states in frustrated magnets with a variety of complex structures, which confine skyrmions to a nanostripe.

We studied minimal-energy states in a stripe of a frustrated magnet (Fig. 1a)[13],

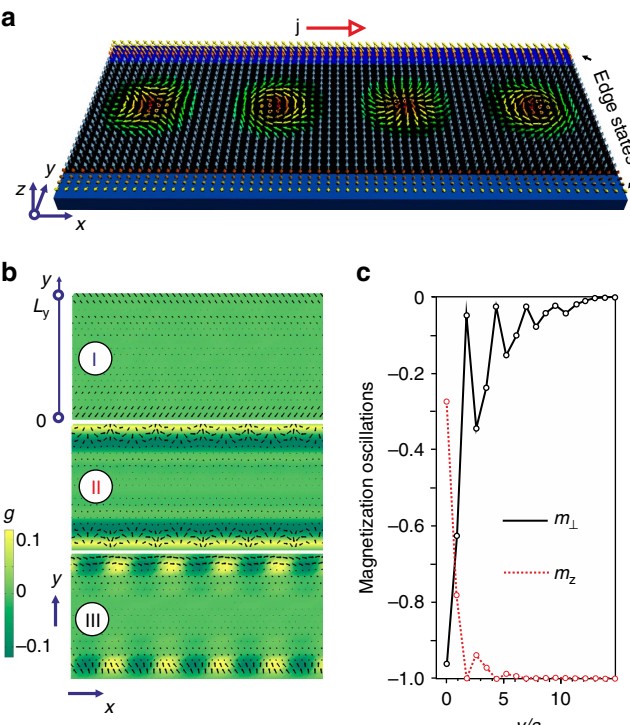

$$E = -J_1 \sum_{\langle i,j \rangle} \mathbf{m}_i \cdot \mathbf{m}_j + J_2 \sum_{\langle\langle i,j \rangle\rangle} \mathbf{m}_i \cdot \mathbf{m}_j - h \sum_i m_i^z$$
$$- \frac{K}{2} \sum_i \left(m_i^z\right)^2 - \frac{1}{2} \sum_i K_i' \left(m_i^z\right)^2, \tag{1}$$

where $\mathbf{m}_i$ is the unit vector in the direction of the magnetization at the site $i$ of a triangular lattice. The first and the second terms in the energy describe the competing ferromagnetic nearest-neighbour and antiferromagnetic next-nearest-neighbour interactions ($J_1, J_2 > 0$), $h$ is the magnetic field applied in the $z$ direction normal to the stripe, $K > 0$ is the bulk easy axis magnetic anisotropy and $K_i' < 0$ is the easy plane anisotropy added near the edges. The rich phase diagram of this model counts 8 different phases including the skyrmion crystal state[13]. Important for our present study is a large region of the field-induced collinear ferromagnetic state, where isolated skyrmions are stable. We use the set of bulk model parameters, $J_2 = 0.5$, $h = 0.4$, $K = 0.2$ (in units of $J_1 = 1$), for which spins inside the stripe are normal to the stripe plane. The easy plane edge anisotropy favors a conical spiral state near the edges, which gives rise to edge states with complex spin structures.

Figure 1b shows three types of evanescent edge states induced by the easy plane surface anisotropy in one or more rows at the

**Figure 1 | Edge states in a frustrated magnet.** (a) Schematic of current-driven motion of skyrmions with various vorticities and helicities confined by the edge states in the stripe of width $L_y$. The electric current runs along the $x$ axis. The magnetic field antiparallel to the $z$ axis induces a saturated ferromagnetic state inside the stripe, in which skyrmions are stable topological defects. The colour of arrows from red ($m_z = -1$) to blue ($m_z = +1$) indicates the alignment of the magnetization vector with respect to the magnetic field. The edge states are induced by changing the type of magnetic anisotropy (from easy axis to easy plane) in one or two rows of spins near the edges of the stripe. (b) Three types of edge states: (I) the state with parallel in-plane spin components, (II) the evanescent conical spiral state and (III) the state with fan-like oscillations of in-plane spins. Arrows show the in-plane components of magnetization; colour indicates the scalar chirality of spin triangles $g = (\mathbf{m}_1 \cdot \mathbf{m}_2 \times \mathbf{m}_3)$ proportional to the topological charge density. (c) Oscillations of the magnetization components, $m_z$ and $m_\perp = \sqrt{1 - (m_z)^2}$, along the $y$ coordinate (dotted red line and solid black line, respectively) in type I edge state near the lower edge of the nanostripe. Distances are measured in units of the lattice constant $a$. The width of the stripe $L_y$ is large enough to avoid an overlap between the upper and lower edge states.

edges of the magnetic stripe. The type I edge state with collinear in-plane spin components is induced by $K' \leq -0.434$ in the first row; $K' = 0$ in the first row and $K' \leq -0.808$ in the second row induces type II state—the evanescent conical spiral state with the wave vector along the boundary ($x$ direction) and the in-plane magnetization vector rotating around the $z$ axis. $K' = -0.406$ in the first row and $K' = -0.203$ in the second row give rise to type III state, in which the in-plane magnetization vector shows fan-like oscillations around a fixed direction in the $xy$ plane. Importantly, in all edge states the in-plane magnetization oscillates with the decaying amplitude along the $y$ axis normal to the edge (Fig. 1c).

These oscillations are a characteristic property of frustrated magnets and the three types of edge states are generic. The origin of the oscillations can be understood by considering asymptotic of the in-plane magnetization vector, $\mathbf{m}_\perp(\mathbf{x}) \propto e^{i\mathbf{q} \cdot \mathbf{x}}$, deep inside the magnetic stripe. In the continuum limit, $|q| \ll 1$,

$$bq^4 - 2aq^2 + K + h = 0, \qquad (2)$$

where the first two terms originate from the expansion of the exchange energy of a frustrated magnet in powers of $q$ ($a = \frac{3}{8}(3J_2 - J_1) > 0$, $b = \frac{3}{64}(9J_2 - J_1) > 0$). This bi-quadratic equation with real coefficients has four solutions: $\pm q = \pm(q' + iq'')$ and $\pm q^\star = \pm(q' - iq'')$. In the situation when modulated states are suppressed in the bulk, all four wave vectors have a nonzero imaginary part $q''$. They can be grouped into two pairs according to the sign of $q''$: $(+q' + iq'', -q' + iq'')$ and $(+q' - iq'', -q' - iq'')$. One pair describes $\mathbf{m}_\perp$ with an amplitude decreasing away from the upper edge and another pair describes the evanescent state near the lower edge. The real parts of the two wave vectors in each pair have opposite signs. The interference between the modulations with positive and negative $q'$ leads to spin oscillations.

In fact, any magnetic defect in frustrated magnets, such as skyrmion or domain wall, gives rise to similar decaying spin oscillations. They lead to sign changes of the skyrmion-skyrmion interaction potential as a function of distance between two skyrmions[13]. Similarly, the interaction energy of skyrmion with an edge state, $U(y)$, obtained by setting $m_z(x, y) = -1$ and minimizing the energy (1) with respect to spins at all other sites, oscillates with $y$ (Fig. 2a). This leads to a sequence of edge channels centred around minima of $U(y)$ (Fig. 2b), which run continuously along the boundaries of a nanostructure and guide the motion of skyrmions.

In equation (2) $q^2 = q_x^2 + q_y^2$. For periodic boundary conditions along the $x$ direction, $q_x = \frac{2\pi N}{L_x}$, where $L_x$ is the length of the stripe and $N$ is an integer number. Thus equation (2) gives $q_y$ for a given $N$. For type I state $N = 0$, for type II state $N = 5$ shown in Fig. 1b and type III state is a superposition of type I and type II states.

Edge channels are closely related to helicity reversals around skyrmions in achiral systems[13,28], which becomes clear if we calculate the potential $U(y, \chi)$ (Fig. 2c) for skyrmion near type I edge state in the following way. In addition to constraining the skyrmion position by $m_z(x, y) = -1$, we constrain its helicity, $\chi$, (ref. 3) by imposing the in-plane spin directions at six sites neighbouring to the skyrmion centre. We also constrain the in-plane spin directions at the edge by $\varphi(\text{edge}) = 0$, where $\varphi$ is the azimuthal angle describing the direction of $\mathbf{m}$. Edge states, like skyrmions, have a helicity and the potential $U(y, \chi)$ depends on the relative helicity of skyrmion and edge state. Figure 2c shows that the helicity angle in the edge channels, corresponding to minima of $U(y, \chi)$, alternates between 0 and $\pi$.

**Motion of skyrmions through the edge channels.** In the continuum limit, in which the period of modulated states and the skyrmion diameter are much larger than the lattice constant, the

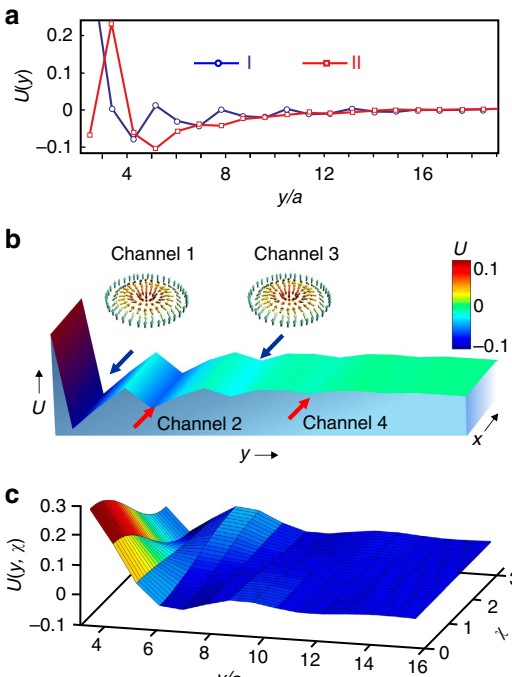

**Figure 2 | Edge channels. (a)** The skyrmion potential energy $U$ (measured in units of $J_1$) versus the distance $y$ between the skyrmion centre and the edge of the stripe for type I (blue line with circular markers) and type II (red line with square markers) edge states. $U(y)$ was calculated by imposing the constraint, $m_z = -1$, at the skyrmion centre and minimizing the energy with respect to spins at all other sites. **(b)** The potential energy landscape for skyrmion interacting with type III state. The local minima of $U(y)$ give rise to a sequence of edge channels separated by the potential barriers, which guide the current-induced motion of skyrmions. The edge channels are numbered starting from the edge (that is, the first channel corresponds to the global minimum of $U(y)$) and are indicated by red and blue arrows. **(c)** The skyrmion potential energy $U(y, \chi)$ calculated by constraining the skyrmion centre, $m_z(x, y) = -1$, and the helicity, $\chi$, of six neighbouring spins (see text for details) for type I edge state. The corresponding 2D plot with a colour bar is shown in Fig. 4a.

spin model (1) can be used to describe itinerant frustrated magnets. We study the current-induced dynamics of skyrmions and antiskyrmions in the stripe of a frustrated magnet with edge states by solving Landau–Lifshitz–Gilbert equation (see 'Methods') with the electric current $j_x$ running along the stripe (Fig. 1a, Supplementary Movies 1 and 2). In contrast to chiral magnets, the skyrmion vorticity $v$ in frustrated magnets can have either sign, $v = \pm 1$ (refs 3,12,13), so that for a given direction of the magnetic field, the skyrmion topological charge, $Q = -\text{sign}(h)v$, also can have either sign. Assuming $h < 0$, we call the magnetic defects with positive vorticity, $v = Q = +1$, skyrmions, while those with $v = Q = -1$ are called antiskyrmions.

Figure 3a shows the time dependence of the $y$ coordinate of skyrmion, initially placed into the channel 3, for several values of the electric current (Supplementary Movie 1). A relatively low current, $j_x = 0.025j_0$, moves skyrmion along the channel, where the unit of current $j_0$ is defined in the 'Methods', equation (9). A larger current, $j_x = 0.05j_0$, forces skyrmion to jump into the channel 2, and $j_x = 0.1j_0$, eventually brings skyrmion into the channel 1 (the one closest to the edge). Figure 3b shows that the $x$ component of the skyrmion velocity $V_x$ varies, when the skyrmion moves across channels, and approaches a constant value, when it moves in a channel. Thus the channel, in which skyrmion moves, can be selected by the electric current.

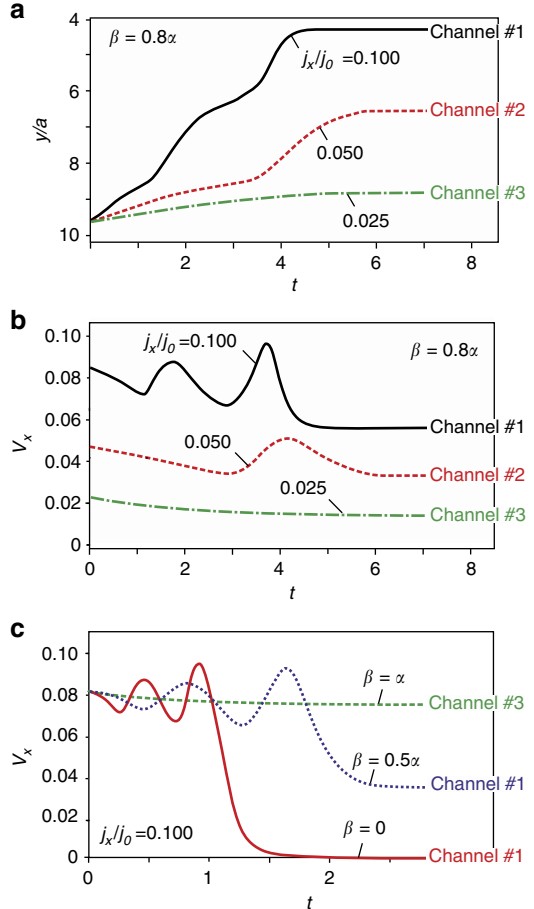

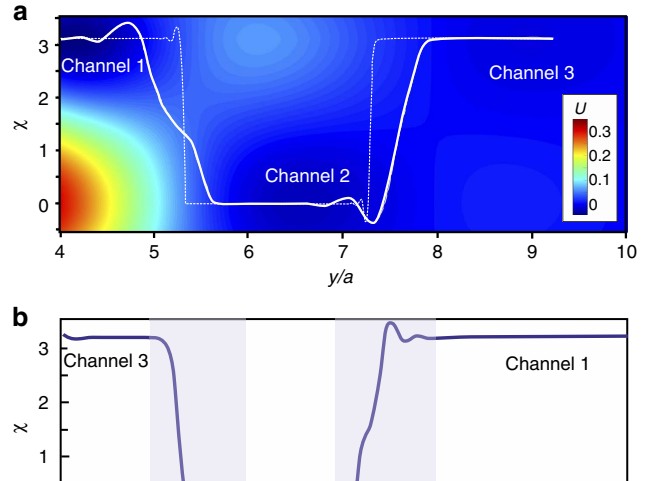

**Figure 4 | Relative helicity of skyrmion and edge state.** (**a**) $y$ dependence of the relative helicity, $\chi$, obtained from the numerical solution of the Landau–Lifshitz–Gilbert equation (solid white line) and from the solution of Thiele equations (3) and (4) (dotted line). The colour plot shows the potential $U(y, \chi)$ (Fig. 2). (**b**) Time dependence of the relative helicity of the skyrmion and edge state for $j_x = 0.1 j_0$ (equation (9)) and $\beta = 0.5\alpha$. The time $t$ is measured in units of $\frac{\hbar}{J_1}$.

**Figure 3 | Skyrmion motion in channel structures formed by edge states.** (**a**) Time dependence of the $y$ coordinate of the skyrmion centre for $\beta = 0.8\alpha$, $\alpha$ and $\beta$ being the Gilbert damping constant and the non-adiabatic spin-torque strength, respectively (Supplementary Movie 1). Skyrmion is initially located in the third channel of type III edge state. For $j_x = 0.025 j_0$ (equation (9)) skyrmion moves in the third channel (green dash-dotted line); for $j_x = 0.05 j_0$, it jumps into the second channel (dashed red line) and for $j_x = 0.1 j_0$ it eventually ends up in the first channel (black solid line). The time $t$ is measured in units of $10^3 \frac{\hbar}{J_1}$. (**b**) Time dependence of the skyrmion drift velocity, $V_x$, measured in units of $10^3 \frac{aJ_1}{\hbar}$ for the same parameters as in (**a**). For $j_x = 0.1 j_0$, skyrmion changes channels twice, which gives rise to two humps in $V_x(t)$. In the steady state $V_x \propto j_x$. (**c**) Time dependence of $V_x$ for $j_x = 0.1 j_0$ and several values of $\beta$ (Supplementary Movie 2). For $\beta = \alpha$ (green dashed line), skyrmion always moves in the third channel; for $\beta = 0.5\alpha$ (blue dotted line), skyrmion jumps into the second channel, in which it moves together with the edge state; for $\beta = 0$ (solid red line), skyrmion ends up in the first channel and comes to a halt. For the steady state motion along a channel, $V_x \propto \beta$.

The driving force for the channel switching is the skyrmion Hall effect, that is, the skyrmion motion with a velocity $V_y$ in the direction transverse to the applied current[29,30]. The skyrmions and antiskyrmions are deflected towards opposite edges of the magnetic stripe, which eventually drives them into the edge channels where both $V_x$ and $V_y$ are strongly affected by the oscillating edge-state potential, $U(y, \chi)$.

The skyrmion motion through the system of edge channels is qualitatively described by Thiele equations[30]:

$$\begin{cases} \alpha \Gamma V_x - G V_y = -\beta \Gamma j_x, \\ G V_x + \alpha \Gamma V_y = -\frac{\partial U}{\partial y} - G j_x, \end{cases} \quad (3)$$

which have to be solved together with the equation for the relative helicity angle (Supplementary Note 1),

$$\mathcal{M} \ddot{\chi} + \alpha \Gamma_{\chi\chi} \dot{\chi} = -\frac{\partial U}{\partial \chi}. \quad (4)$$

Here $G = \int d^2 r\, \mathbf{m} \cdot \frac{\partial \mathbf{m}}{\partial x} \times \frac{\partial \mathbf{m}}{\partial y} = 4\pi Q$, $\Gamma = \int d^2 r \frac{\partial \mathbf{m}}{\partial x} \cdot \frac{\partial \mathbf{m}}{\partial x} = \int d^2 r \frac{\partial \mathbf{m}}{\partial y} \cdot \frac{\partial \mathbf{m}}{\partial y}$, $\Gamma_{\chi\chi} = \int d^2 r (1 - (m^z)^2)$, $\mathcal{M}$ is the 'helicity mass', $\alpha$ is the Gilbert damping constant and $\beta$ describes the non-adiabatic spin-torque.

For an unconstrained skyrmion motion away from the edges ($U = 0$) and equation (3) gives

$$V_x \approx -j_x, \quad V_y \approx -(\alpha - \beta)\frac{\Gamma}{G} j_x, \quad (5)$$

for $\alpha, \beta \ll 1$. On the other hand, for a skyrmion moving along the channel ($V_y = 0$),

$$V_x = -\frac{\beta}{\alpha} j_x, \quad V_y = 0 \quad (6)$$

and $F_y = -\frac{\partial U}{\partial y} = \frac{(\alpha - \beta)}{\alpha} G j_x$.

The skyrmion Hall effect pushes skyrmion towards the edge and when it moves from one channel into another the helicity angle varies by $\pm\pi$ (Fig. 4a,b), which explains the rotation of spins in the skyrmion that occurs during the channel switching (Supplementary Movie 2). The solution of equations (3) and (4) reproduces the jumps in the helicity angle (dotted line in Fig. 4a) and is in reasonable agreement with the skyrmion trajectory extracted from the numerical solution of the Landau–Lifshitz–Gilbert equation (solid line in Fig. 4a). The only fitting parameter of the simplified description of skyrmion dynamics is the helicity mass $\mathcal{M}$.

Equation (6) also applies to type II and type III edge states that are inhomogeneous along the $x$ axis and, therefore, move along the boundary when an electric current is applied. When skyrmion is captured by such an edge state, they move with the same speed.

This speed is proportional to $\beta$ and the motion stops for $\beta = 0$ (Fig. 3c, Supplementary Movie 2).

The simultaneous presense of skyrmions and antiskyrmions in frustrated magnets can be very useful for implementation of logical operations. However, since skyrmions and antiskyrmions have the same energy, they can form random mixtures (Fig. 5a).

The skyrmion Hall effect can be employed to separate them: since the sign of the transverse velocity depends on the sign of topological charge (equation (5)), skyrmions and antiskyrmions under an applied current move towards opposite edges. Supplementary Movie 3 shows how the skyrmion Hall effect and a notch at one of the edges help to filter out antiskyrmions. The final state with skyrmions separated from antiskyrmions is shown in Fig. 5b.

**Instability of edge states.** Above a critical electric current edge states become unstable against emission of skyrmion–antiskyrmion pairs. Figure 6 and Supplementary Movie 4 show complex dynamics of type II edge states under an applied current $j_x = 0.02 j_0$. These spiral edge states with in-plane spins at the edge are characterized by the winding number, $N = \pm \frac{(\varphi(L_x) - \varphi(0))}{2\pi}$, where $\varphi$ is the angle describing the spin orientation and the $+/-$ sign is for the upper/lower edge. For periodic boundary conditions along the $x$ axis, $N$ is an integer number. In the initial state (Fig. 6a) $N_u = +1$ at the upper edge and $N_d = -3$ at the lower edge. Under the applied current the upper-edge state becomes unstable and emits two skyrmion–antiskyrmion pairs (Fig. 6b,c). The skyrmion Hall effect pushes skyrmions towards the lower edge, while antiskyrmions return to the upper edge (Fig. 6d,e). After the skyrmions and antiskyrmions have vanished at the corresponding edges, the winding numbers of the edge states become $N_u = N_d = -1$ (Fig. 6f). Then the lower edge state becomes unstable and emits two skyrmion–antiskyrmion pairs (Fig. 6g), which separate into two skyrmions that return to the lower edge and two skyrmions that move towards the upper edge (Fig. 6h). In the final state (Fig. 6i) the winding numbers of the edge states are $N_u = -3$ and $N_d = +1$, that is, in the end of these transformations the upper and lower edge states exchanged their winding numbers. The final state is stable.

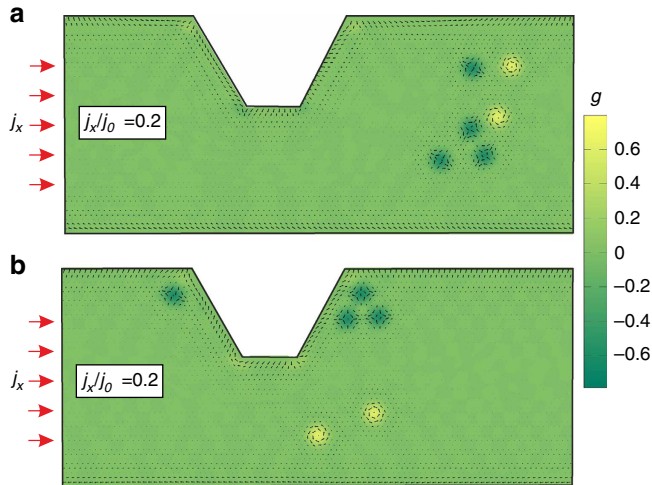

**Figure 5 | Separation of skyrmions from antiskyrmions in a nanotrack with a notch.** (**a**) The initial state of two skyrmions (their positive topological charge density is indicated by yellow colour) and four antiskyrmions (dark green colour indicating negative topological charge density). (**b**) The final configuration after the electric current $j_x/j_0 = 0.02$ (equation (9)) was applied for the time $t = 1.8 \times 10^4 \frac{\hbar}{J_1}$ (Supplementary Movie 3). The motion of skyrmions is hindered by a notch. Skyrmions are deflected towards the lower-edge channels of the nanotrack.

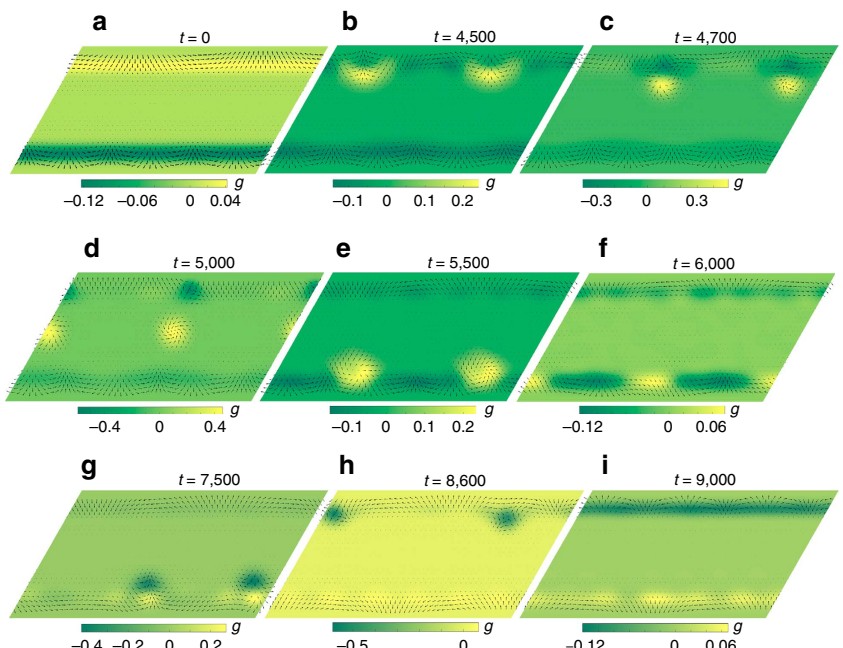

**Figure 6 | Instability of edge states.** (**a**) The initial state with the winding numbers $N_u = +1$ and $N_d = -3$. (**b–d**) Emission of two skyrmion–antiskyrmion pairs from the upper edge. Antiskyrmions are driven by the skyrmion Hall effect towards the upper edge. When they are absorbed the winding number of the upper edge state changes to $N_u = -1$. (**e**) Two skyrmions are absorbed by the lower edge state and change its winding number to $N_d = -1$. (**f,g**) Emission of two skyrmion–antiskyrmion pairs from the lower edge. Two skyrmions return to the lower edge and change its winding number to $N_d = +1$. (**h**) Two antiskyrmions pass through the upper edge, after which $N_u = -3$. (**i**) The final state with the interchanged winding numbers is stable. This simulation was performed for $j_x = 0.02 j_0$ (equation (9)), $\beta = 0$, $J_2 = 0.36 J_1$ and $h = 0.05 J_1$. The time $t$ is measured in units of $\frac{\hbar}{J_1}$.

These metamorphoses can be understood, if we notice that the type II edge state with in-plane spins at the edge has topological charge, $Q_{edge} = \frac{1}{2}(m^z_{edge} - m^z_{bulk})N$, where $m^z_{edge}$ is the $z$-component of the unit magnetization vector $\mathbf{m}$ at the edge, $m^z_{bulk}$ is the bulk value of $m^z$ and $N$ is the winding number of the edge state. For $m^z_{edge} = 0$ and $m^z_{bulk} = -1$, the topological charge, $Q_{edge} = \frac{1}{2}N$, is integer or half-integer. More generally, edge states carry a fractional topological charge. The reason for the instability of the upper-edge state, which initially had $Q_u = +1/2$, is the skyrmion Hall effect that, for $j_x > 0$, pushes this state downwards. Similarly, the lower-edge state with a negative topological number is pushed upwards, which leads to its instability (Fig. 6f,g). In the stable final state (Fig. 6i) the upper-edge state has negative topological charge, $Q_u = -3/2$, and lower-edge state has positive topological charge, $Q_d = +1/2$. These states can be made unstable by reversing the direction of the electric current.

Remarkably, when skyrmion or antiskyrmion vanishes at an edge, the winding number of the edge state changes by $\pm 1$, which suggests that such processes are governed by a conservation law of topological nature. The total topological charge of the stripe equal the sum of topological charges of skyrmions, antiskyrmions and the two edge states, is not conserved. For example, when a skyrmion with $Q = +1$ passes through the lower edge, the topological charge of the edge state increases by $+1/2$. What is conserved, is the total vorticity equal the sum of vorticities of skyrmions and antiskyrmions inside the stripe and the winding numbers of the edge states:

$$\nu_{total} = N_s - N_a + N_u + N_d. \tag{7}$$

Here, $N_s$ is the number of skyrmions with vorticity $+1$ and $N_a$ is the number of antiskyrmions with vorticity $-1$. One can check that $\nu_{total}$ is invariant under all transmutations shown in Fig. 6, including the emission of skyrmion–antiskyrmion pairs. When skyrmion is absorbed by an edge, the ring of in-plane spins winding around the skyrmion centre is cut, stretched into a straight segment and becomes a part of the edge state, which explains the conservation of vorticity.

The current-induced instability of edge states in frustrated magnets that leads to emission of skyrmion–antiskyrmion pairs, is analogous to the magnetic-field-induced instability of chiral magnets, which generates chains of skyrmions parallel to the edges[31–33]. The novel aspect of our study is the crucial role of the edge state topology and the skyrmion Hall effect for the emergence of the instability. The skyrmion Hall effect is likely involved in the nucleation of skyrmions at boundaries with sharp corners[30]. Equation (7) describing the conservation of the total vorticity in the skyrmion/antiskyrmion absorption by the edge states is similar to the winding number conservation that governs the dynamics of domain walls and half-integer vortices in ferromagnetic nanostripes[34,35]. In those systems, however, magnetization is confined to the stripe plane, whereas in our case the magnetization inside the stripe is vertical.

## Discussion

In conclusion, we showed that the states formed at the edges of frustrated magnets give rise to interesting physics, which can be useful in more than one way. The multiple edge channels continuously running along the boundaries of magnetic nanostructures and guiding the motion of skyrmions can be employed for magnetic patterning of nanodevices. Skyrmions, which fit perfectly into these edge channels, can be directed by pulsed currents along different paths. The simultaneous presence of skyrmions and antiskyrmions, which under an applied current move towards opposite edges, opens additional possibilities to do logical operations with these topological objects. Our results suggest that information can be stored in winding numbers of edge states and manipulated by electric currents through the exchange of skyrmions and antiskyrmions between the edges. These results open new avenues for design of magnetic devices.

## Methods

**Landau–Lifshitz–Gilbert equation.** The current-driven dynamics of spin textures was simulated using Landau–Lifshitz–Gilbert equation for the unit vector $\mathbf{m}$ in the direction of magnetization,

$$\left(1 + \alpha^2\right)\frac{\partial \mathbf{m}}{\partial t} = -\left(\mathbf{m} \times \mathbf{H}_{eff} + \alpha \mathbf{m} \times [\mathbf{m} \times \mathbf{H}_{eff}]\right) + (1 + \alpha\beta)(\mathbf{j} \cdot \nabla)\mathbf{m}$$
$$+ (\alpha - \beta)[\mathbf{m} \times (\mathbf{j} \cdot \nabla)\mathbf{m}], \tag{8}$$

which was solved using fourth-order Runge–Kutta method. Here, $\mathbf{H}_{eff}$ is a local effective magnetic field, which at the site $i$ is given by $\mathbf{H}_{eff} = -\partial E/\partial \mathbf{m}_i$, $\alpha = 0.01$ is the Gilbert damping constant and $\beta$ is the dimensionless strength of the non-adiabatic torque[36,37]. All physical quantities in our calculations are dimensionless: we measure time $t$ in units of $\frac{\hbar}{J_1}$, where $J_1$ is the nearest-neighbor exchange constant, energy $E$ in units of $J_1$, distances in units of the triangular lattice constant $a$, current density in units of

$$j_0 = \frac{2eJ_1 a}{\hbar p v_0}, \tag{9}$$

where $e$ is the absolute value of the electron charge, $p$ is the spin polarization of the electric current and $v_0$ is the unit cell volume (we assume that the unit cell contains one spin $S = 1$). Spin configurations were first relaxed at zero current. When the convergence was reached, the electric current was applied along the $x$ direction.

**Data availability.** The data that support the findings of this study are available from the corresponding author on request.

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

## Acknowledgements

The authors would like to thank N. Nagaosa for interesting discussions. This study was supported by the FOM Grant 11PR2928.

## Author contributions

All authors contributed to all aspects of this work.

## Additional information

**Competing financial interests:** The authors declare no competing financial interests.

**Publisher's note**: 

