## [Peer Review File · Nature Communications]

Reviewers' comments:

Reviewer #1 (Remarks to the Author):

In the manuscript, the authors studied the skyrmion dynamics in a nanostripe of inversion symmetric magnets. The research field of skyrmion has expanded significantly since the experimental observation of this topological excitation in 2009. The dynamics of skyrmion in confined geometry has been studied extensively for skyrmions in chiral magnets, partly because of the applications of skyrmion in racetrack memory devices. Here the authors considered the inversion symmetric magnets which can also host skyrmions. As discussed in the manuscript and also other published work, the skyrmions in inversion symmetric magnets have unusual properties which are not shared by skyrmions in chiral magnets. By exploiting these unusual properties, the authors found some interesting consequences for skyrmions in a nanostripe, which could have potential for applications. The main achievements are:

1. The authors found several channels for the skyrmions moving along the edge. These channels correspond to the energy minima in the skyrmion potential created by the edges.
2. The authors revealed an instability of edge states in the presence of a current. The instability leads to the skyrmion creation.

In my opinion, there are also a few loose ends:

1. The physics discussed in the manuscript depends on the boundary condition. The authors have assumed an easy plane anisotropy at the edge, without explaining why. Moreover, the edge region where the surface anisotropy is operational is also loosely defined. The authors need to justify why the boundary condition used in their calculations.
2. The Thiele's equation of motion in Eq. (3) only describes the Goldstone mode associated with the translational motion of skyrmion. As the authors have discussed, there exists gapless helicity mode. In this case, Eq. (3) needs to be revised in order to account for the both Goldstone modes.
3. The authors calculated the surface potential for a skyrmion near edge. No details on such calculations are given. There may exist preferred channels for skyrmions to move along the edge which can be regarded as some sort of energy minima in the skyrmion surface potential. However, I wonder if one can treat skyrmion as a particle-like object when it approaches the boundary. The skyrmion is distorted and the position of the skyrmion is not well defined.

With these regards, I do not think the present manuscript meets the high standard of Nature Communications. Especially, without a detailed and careful discussion on the boundary condition, it is unclear how the present numerical simulations are related to experiments. I suggest that the paper is published in a different journal.

A few more technical comments/questions:

1. What is the definition of the topological charge at the edges? Why it is given by $Q_{\text{edge}} = (m_{\text{edge}}^z - m_{\text{bulk}}^z) N/2$?
2. Should it be all "plus" sign in Eq. (6)?

Reviewer #2 (Remarks to the Author):

On the basis of the recent finding that certain frustrated magnets could sustain novel type of skyrmions together with antiskyrmions in contrast to the standard chiral magnets, the authors theoretically investigated the current-driven dynamics of such frustration-induced skyrmions (antiskyrmions) in more or less realistic situation of nanostripes with edges. The study is based on the same authors' previous Nature Commun. results and on the model analyzed there, but is much extended in the direction of novel edge states and current-induced skyrmion (antiskyrmion) dynamics, including the complex edge states, a way of separating skyrmions from antiskyrmions,

and emission of skyrmion-antiskyrmion pairs from the edges, etc. The results are highly interesting, and might provide a theoretical basis for the future technology of skyrmion manipulations. As such, the referee is basically quite positive about its publication in Nature Commun.

Meanwhile, the referee finds several points to be further detailed or corrected, which are given below (mostly minor ones). The referee recommends the publication of the article after the authors have taken care of these points.

=====

=1. On page 3, the second last paragraph, the explanation of the K' -parameter assignment sounds a bit ambiguous for the type II and III configurations. In type II, the authors mention the K' condition on the second row only, but what about K' on the first row? In type III, the authors mention "in the first two rows". What does "two" mean?

2. In Figs.1(a) and (b), "L" in the figure might be better "Ly" to be consistent with the caption. In Fig.1(c), the line mark for "mz" in the legend seems to be missing. It is not very clear in Fig.1(c) which range of Fig.1(b) its abscissa corresponds to. The authors might give the explicit Ly-value in the figure caption so that the reader could judge the meaning of the numbers on the abscissa of Fig.1(c).

3. The referee guesses that the coefficients "a" and "b" in eq.(2) can be written in terms of the parameters J1 and J2 in the energy expression. eq. (1). If so, please provide these expressions explicitly.

4. On the second last line of page 5, what "itinerant" means? The word usually means "conducting" or "metallic", but is it the case here? How this is related to "the continuum limit"?

5. In the caption of Fig.3(a), the parameters α and β are given without their definitions. In the main text, Fig.3(a) is referred to before these parameters are first defined in the text, i.e., after eq.(3) below. The authors might add a statement about α and β in the caption of Fig.3(a).

6. On the second and the third lines of page 7, "Fig.5h" and "Fig.5e" should be "Fig.5g" and "Fig.5h"?

=====

== < BR >

The references and figure/equation numbers in this reply pertain to the revised version of the manuscript. The changes made in the manuscript are marked by blue colour.

Reviewer #1:

1. *The physics discussed in the manuscript depends on the boundary condition. The authors have assumed an easy plane anisotropy at the edge, without explaining why. The authors need to justify why the boundary condition used in their calculations. Moreover, the edge region where the surface anisotropy is operational is also loosely defined. The authors need to justify why the boundary condition used in their calculations.*

We thank the reviewer for this question. Edge states, which are necessary to keep skyrmions within nanotracks, are indeed strongly influenced by boundary conditions or, more precisely, by spin interactions at the edges, which may differ from those in the bulk. In the literature on chiral skyrmions edge interactions are often neglected. The reason for that might be that in chiral magnets edge states are also induced by the *bulk* Dzyaloshinskii-Moriya (DM) interactions. In absence of inversion symmetry, the DM interactions give rise to the so-called Lifshitz invariants proportional to the gradient of magnetization, which result in a twisting of spins near the edges [21-24]. In frustrated magnets considered in our manuscript Lifshitz invariants are forbidden by symmetry. Therefore, bulk interactions in centrosymmetric skyrmion materials do not favor edge states. Such states can still be stabilized by surface or interfacial magnetic anisotropies as well as by other spin interactions at the edges which, in principle, should also be taken into account in chiral magnets.

In this work we focused on effects of surface magnetic anisotropy resulting from symmetry lowering at the edges (references [25-27] cited in the manuscript and L. Neel, J. Phys. Rad. **15**, 225 (1954); A. Hubert, R. Schaefer, Magnetic Domains, Springer, Berlin, 1998, p. 107). An *easy axis* surface anisotropy would lead to further stabilization of the collinear field-induced magnetic state and would not give rise to edge states. On the other hand, an *easy plane* magnetic anisotropy favors in-plane spins at the edges and can stabilize (conical) spiral states at the edges (as is clear from the phase diagram of the frustrated triangular magnet, Fig. 1 in Ref. 5), which is our rationale for considering the easy plane surface anisotropy.

Another important point is that our conclusions concerning the formation of edge channels and topology of edge states are generic. Although the edge state type (see Fig. 1) does depend

on details of the spin lattice Hamiltonian in a few rows close to the edge, the asymptotic form of the edge state inside the nanostripe and, in particular, spin oscillations giving rise to edge channels are universal. Equation (2) shows that these spin oscillations follow from the expansion of the *bulk* spin Hamiltonian in powers of the wave vector \mathbf{q} of the spin modulation, bulk magnetic anisotropy and the applied magnetic field. The edge state topology (the winding number of the edge state) is determined by q_x - the component of the wave vector along the stripe direction.

Different choices of edge anisotropy considered in our manuscript were used to identify possible edge states. The three types that we have found in our simulations are shown in Fig. 1. These choices are not unique: the spiral (type II) edge state shown in Fig. 1b is stabilized by anisotropy in the second layer, while the spiral states shown in Fig. 5 were obtained for anisotropy of equal strength in three surface layers. Such choices are also not unphysical. For instance, a magnetic anisotropy in several surface layers was used to explain the magnetic patterns found by the resonant x-ray scattering on the chiral magnet Cu_2OSeO_3 [S. L. Zhang et al., Multidomain Skyrmion Lattice State in Cu_2OSeO_3 , *Nano Lett.* **16**, 3285 (2016)]. The surface anisotropy that gradually decays into the bulk was discussed in A.N. Bogdanov, U.K. Roessler, K.-H. Mueller, Magnetic anisotropy, phase transitions, and domain structures in films with out-of-plane magnetization. *J. Magn. Magn. Mat.* **238**, 155–159 (2002).

The referee notes that the edge region in our simulations is not well defined. We think that it is defined much better than the rough edges of realistic nanostructures. From theoretical perspective it is important that the positions of all edge channels lie outside the region where the magnetic anisotropy is modified.

Finally, we would like to point out that the topological nature of the edge states renders them locally stable even if they do not minimize the total energy of a nanostripe. This metastability is important for the control of the topological number of edge states by emission or absorption of skyrmions, discussed in the last part of the manuscript.

In fact, many of the points of this reply have already been discussed (shortly) in the first version of our manuscript:

“...Therefore, the practical use of skyrmions crucially depends on their repulsion from edges of magnetic nanostructures. In chiral magnets such a repulsion is naturally provided by the bulk DM interaction, which tilts the magnetization vector away from the magnetic field direc-

tion at the edges of a magnet, giving rise to the so-called edge states \cite{Wilson2013,Rohart2013, Monchesky2014,Keesman2015}.

Competing spin interactions in frustrated magnets do not necessarily induce similar spin tilts. However, exchange interactions and magnetic anisotropies at surfaces or interfaces of magnetic materials can be significantly different from those in bulk, because of a lower symmetry of magnetic ions at the edges \cite{Gay1986, Bruno1989, Johnson1996}. We have found that a strong surface anisotropy gives rise to edge states in frustrated magnets with a variety of complex structures, which confine skyrmions to a nanostripe.’’

In response to this comment we have made the following changes in the text:

*The end of the paragraph below Eq.(1): We use the set of bulk model parameters, $J_2 = 0.5$, $h = 0.4$, $K=0.2$ (in units of $J_1 = 1$), for which spins *inside the stripe* are normal to the stripe plane. The easy plane edge anisotropy favors a conical spiral state near the edges, which gives rise to edge states with complex spin structures.*

2. The Thiele's equation of motion in Eq. (3) only describes the Goldstone mode associated with the translational motion of skyrmion. As the authors have discussed, there exists gapless helicity mode. In this case, Eq. (3) needs to be revised in order to account for the both Goldstone modes.

This is a very good point and we thank the referee for bringing it up. We have observed the skyrmion rotation (helicity dynamics) in our simulations, but since it only took place when the skyrmion was switching channels, we thought that it was caused by a deformation of skyrmion moving through a potential barrier and that it played a minor role, which is why we did not include helicity into Thiele equations of motion.

In fact, the global spin rotation around the z axis (the Goldstone mode mentioned by the referee) varies very little and is not very important. On the other hand, the relative helicity of the skyrmion and the edge state, $\chi = \chi_{Skyrmion} - \chi_{Edge}$, shows large changes as skyrmion moves from one channel to another (since spins in the edge state have in-plane components, the edge state also has a helicity, χ_{Edge}). The relative helicity is not a Goldstone mode, as the interaction between the skyrmion and the edge state strongly depends on it.

The potential $U(y, \chi)$ is shown in Fig. 2c. The minima of this potential corresponding to the edge channels lie alternately at the lines $\chi = 0$ and $\chi = \pi$, e.g., the minimum that gives rise to the first and third channels is at $\chi = \pi$, while the second-channel minimum is at $\chi = 0$. This alternation reflects the helicity reversals in non-chiral systems [ref 13 and Yu, X. Z. et al. Magnetic stripes and skyrmions with helicity reversals. Proc. Natl Acad. Sci. USA} **109**, 8856-8860 (2012)]. The potential shown in the earlier version of our manuscript (which is now called $U(y)$ and is shown in Fig. 2a) is obtained by minimizing $U(y, \chi)$ with respect to χ at a given y (this also answers the question 3 of the reviewer 1). Although the positions of minima of the potential shown in Fig. 2a coincide with the centers of the edge channels, it does not provide full information necessary to understand the skyrmion dynamics. When skyrmion jumps from one channel into another, the angle χ varies by $\pm \pi$, which is the real reason for the observed rotations of skyrmion spins (see Fig. 4a,b). At the same time, our conclusions concerning stable positions of skyrmions within the edge channels (if the applied electric current is not large enough to overcome the potential barrier) remain unchanged, since at such points $\frac{\partial U}{\partial y} = \frac{\partial U}{\partial \chi} = 0$, which justifies the use of the potential minimized with respect to χ .

We explain these points in the resubmitted manuscript and provide a simplified description of the dynamics found in numerical simulations using the coupled Thiele equations for the skyrmion center-of-mass coordinates, x and y , and the relative helicity, χ . The derivation of equations for x , y , and χ can be found in the Supplementary information.

Changes made:

We added Eq.(4) describing the helicity dynamics and modified the text around it accordingly. We added a figure (now Fig. 4) describing the skyrmion trajectory in the (y, χ) -space (Fig. 4a) as well as the time dependence of the relative helicity (Fig. 4b). In Fig. 4a we also show the trajectory calculated by solving the Thiele equations for the coupled dynamics, Eqs. (3) and (4).

We discuss the helicity jumps in the text below Eq. (6):

The Skyrmion Hall effect pushes skyrmion towards the edge and when it moves from one channel into another the helicity angle varies by $\pm \pi$, which explains the rotation of spins in the skyrmion that occurs during the channel switching (see Supplementary Movie 2). The solution of the Eqs. (3) and (4) reproduces the jumps in the helicity angle (dotted line in Fig.

4a) and is in reasonable agreement with the skyrmion trajectory extracted from the numerical solution of the Landau-Lifshits-Gilbert equation (solid line in Fig. 4a). The only fitting parameter of the simplified description of skyrmion dynamics is the “mass” M .

3. The authors calculated the surface potential for a skyrmion near edge. No details on such calculations are given. There may exist preferred channels for skyrmions to move along the edge which can be regarded as some sort of energy minima in the skyrmion surface potential. However, I wonder if one can treat skyrmion as a particle-like object when it approaches the boundary. The skyrmion is distorted and the position of the skyrmion is not well defined.

The first part of the question is already answered (see our reply to question 2). Describing skyrmion by its center-of-mass coordinate and helicity and neglecting its deformation is certainly an approximation and the solution of the Landau-Lifshitz-Gilbert (LLG) equation for an array of many spins is, in principle, sufficient. However, we believe that the simplified description of skyrmions with a few collective coordinates gives valuable insights into dynamics of these topological defects and helps to develop physical intuition on how they can be controlled with applied currents and fields.

The potential $U(y, \chi)$ was calculated by minimizing the total spin energy with the constraints $m_z(x, y) = -1$, $\chi_{Edge} = 0$ and by imposing spin directions at the six sites surrounding the skyrmion center to obtain the skyrmion with the helicity χ . Clearly, this simple procedure does not provide full description of complex spin textures found in numerical simulations. However, the minima of $U(y, \chi)$ coincide with the centers of the edge channels deduced from the solution of the LLG equation, the height of the potential barriers determines which channel skyrmion can reach and the peculiar χ -dependence of U explains the observed jumps of the relative helicity angle by $\pm\pi$. Without $U(y, \chi)$ we would not be able to understand the results of our numerical simulations.

In the end of the section “Edge states and edge channels in frustrated magnets” we added the paragraph describing the helicity dependence of the interaction between skyrmion and edge state:

Edge channels are closely related to ‘helicity reversals’ around skyrmions in achiral systems [13,28], which becomes clear if we calculate the potential $U(y, \chi)$ (see Fig. 1a) for skyrmion near the type I edge state in the following way. In addition to constraining the skyrmion posi-

tion by $m_z(x, y) = -1$, we constrain its helicity, χ [3], by imposing the in-plane spin directions at six sites neighboring to the skyrmion center. We also constrain the in-plane spin directions at the edge by $\varphi = 0$, where φ is the azimuthal angle describing the direction of \mathbf{m} , because edge states, like skyrmions, have a helicity and the potential $U(y, \chi)$ depends on the relative helicity of skyrmion and edge state. Figure 1a shows that the helicity angle in the edge channels, corresponding to minima of $U(y, \chi)$, alternates between 0 and π .

We plotted the potential $U(y, \chi)$ in Fig. 2c.

A few more technical comments/questions:

1. What is the definition of the topological charge at the edges? Why it is given by $Q_{\text{edge}} = (m_{\text{edge}}^z - m_{\text{bulk}}^z) N/2$?

First we note that the topological charge,

$$Q = \frac{1}{4\pi} \int d^2x \left(\mathbf{m} \cdot \frac{\partial \mathbf{m}}{\partial x} \times \frac{\partial \mathbf{m}}{\partial y} \right),$$

is only integer, if \mathbf{m} does not change along the boundary of the spin system. This condition does not hold, for example, for magnetic vortices in nanodiscs with the in-plane spins rotating along the boundary, which is why the vortices have a half-integer skyrmion charge. The expression for the topological charge can be written in the form,

$$Q = \frac{1}{4\pi} \int d^2x \sin \theta \left(\frac{\partial \theta}{\partial x} \frac{\partial \varphi}{\partial y} - \frac{\partial \theta}{\partial y} \frac{\partial \varphi}{\partial x} \right),$$

where θ and φ are the polar angles describing the direction of the unit vector \mathbf{m} . For the edge state $\theta = \theta(y)$ and $\varphi(L_x, y) - \varphi(0, y) = \pm 2\pi N$, for all y , where N is the winding number of the state and the \pm sign is for the upper/lower edge. Hence,

$$Q = \frac{N}{2} (m_z(\text{edge}) - m_z(\text{bulk})).$$

Now, $m_z(\text{bulk}) = -1$ and for a strong easy plane edge anisotropy $m_z(\text{edge}) = 0$, so that $Q = \frac{N}{2}$. For $m_z(\text{edge}) \neq 0$, the topological charge of the edge state is neither integer nor half-integer, which however is not important for the observed instability of edge states resulting from the skyrmion Hall effect.

2. Should it be all "plus" sign in Eq. (6)? (Now Eq.(7))

The sign in front of N_a is correct, but the sign in front of N_d should indeed be reversed. We thank the referee for pointing this out. The minus sign in front of the number of antiskyrmions, N_a , is due to the negative vorticity of antiskyrmion. For example, if a skyrmion-antiskyrmion pair is emitted from an edge and both N_s and N_a increase by 1, the total vorticity remains unchanged. The + sign in front of the winding number of the lower edge state, N_d , is due to our definition of the winding numbers. If the spins rotate anticlockwise n times along the upper edge, then its winding number is $N_u = n$. If the same happens at the lower edge, then its winding number is $N_d = -n$. With these definitions the topological charges of the upper and lower states are given by $Q_u = N_u/2$ and $Q_d = N_d/2$, respectively. One can check that the number in the expression for the conserved total vorticity are correct by following the changes of N_s, N_a, N_u and N_d in the processes of emission and absorption of skyrmions (see Fig. 5). Initially, the $N_u = +1$ and $N_d = -3$. These winding numbers do not change with two skyrmion-antiskyrmion pairs are emitted by the upper edge. After absorption of two antiskyrmions at the upper edge $N_u = -1$. After absorption of two skyrmions at the lower edge $N_d = -1$, etc.

We corrected the sign in Eq.(7).

Reviewer #2:

1. On page 3, the second last paragraph, the explanation of the K' -parameter assignment sounds a bit ambiguous for the type II and III configurations. In type II, the authors mention the K' condition on the second row only, but what about K' on the first row? In type III, the authors mention "in the first two rows". What does "two" mean?

We thank the referee for this remark. The type II state shown in Fig. 1b by modifying the anisotropy only in the second row. This is by no means a unique way to stabilize type II state. The spiral states shown in Fig. 5 were obtained for an easy plane anisotropy of equal strength in three edge rows. We played with boundary conditions to find out what kind of edge states the nanostructure can, in principle, have. See also our response to referee 1.

As to the type III state, there was indeed a typo in the previous version of the text, which is now corrected.

The description of the surface anisotropies used to produce the three types of edge states now reads:

The type I edge state with collinear in-plane spin components is induced by $K'_x = 0.434$ in the first row; $K'_y = 0$ in the first row and $K'_z = 0.808$ in the second row induces type II state -- the evanescent conical spiral state with the wave vector along the boundary (x -direction) and the in-plane magnetization vector rotating around the z axis. $K'_x = -0.406$ in the first row and $K'_z = -0.203$ in the second row give rise to type III state, in which the in-plane magnetization vector shows fan-like oscillations around a fixed direction in the xy -plane.

2. In Figs.1(a) and (b), "L" in the figure might be better "Ly" to be consistent with the caption. In Fig.1(c), the line mark for "mz" in the legend seems to be missing. It is not very clear in Fig.1(c) which range of Fig.1(b) its abscissa corresponds to. The authors might give the explicit Ly-value in the figure caption so that the reader could judge the meaning of the numbers on the abscissa of Fig.1(c).

We have corrected the typos in Fig. 1 a,b and modified the line mark in Fig. 1c, as the referee suggested. The precise value of L_y is not very informative, because the distance between neighbouring rows in the y -direction is $\sqrt{3}/2$ (the distance between neighboring spins is 1).

Figure 1(c) shows the lower edge state. The width of the stripe is much larger than the spatial extent of the edge state in the y -direction to avoid the overlap between the upper and lower edge states (if that is what the referee is worried about).

We have corrected the typos in Fig. 1 a,b and modified the line mark in Fig. 1c. Corresponding corrections have been made in the caption to Fig. 1.

3. The referee guesses that the coefficients "a" and "b" in eq.(2) can be written in terms of the parameters J_1 and J_2 in the energy expression. eq. (1). If so, please provide these expressions explicitly.

$$a = \frac{3}{8}(3J_1 - J_2), b = \frac{3}{64}(9J_1 - J_2)$$

Incommensurate states are stable for $J_2/J_1 > 1/3$ [13], in which case $a, b > 0$.

Changes made in the text:

Below Eq.(2):

... where the first two terms originate from the expansion of the exchange energy of a frustrated magnet in powers of q ($a = \frac{3}{8}(3J_1 - J_2) > 0, b = \frac{3}{64}(9J_1 - J_2) > 0$).

4. On the second last line of page 5, what "itinerant" means ? The word usually means "conducting" or "metallic", but is it the case here ? How this is related to "the continuum limit" ?

Indeed, itinerant means conducting or metallic. The model Eq.(1) describes a frustrated magnetic insulator with localized spins. Magnetic frustration is, however, not limited to insulators. There are quite a few conducting materials showing non-collinear spin orders, such as the cubic perovskite SrFeO_3 and elemental rare earth metals. Some of them may host skyrmions. In the continuum limit, i.e. when the period of the spiral or the skyrmion diameter are much larger than the lattice constant, our model can be used to describe conducting frustrated magnets. Importantly, Eq. (2) valid in the continuum limit implies that edge channels must also exist in frustrated magnetic conductors. To summarize, Eq. (1) can be thought of as a discretization of a continuum model of magnetically frustrated conductors. In this respect, our approach is not different from the widely used spin lattice model description of itinerant chiral magnets, such as MnSi [see e.g. ref. 30]. The diameter of skyrmions in our simulations is 6 and 10 lattice constants (Figs. 4 and 5, respectively).

5. In the caption of Fig. 3(a), the parameter α and β are given without their definitions. In the main text, Fig.3(a) is referred to before these parameters are first defined in the

text, i.e., after eq.(3) below. The authors might add a statement about α and β in the caption of Fig.3(a).

We thank the reviewer for this remark.

The new caption of Fig. 3a reads: Time dependence of the y-coordinate of the skyrmion centre for $\beta = 0.8\alpha$, α and β being the Gilbert damping constant and the non-adiabatic spin-torque strength, respectively (see Supplementary Movie 1).

6. On the second and the third lines of page 7, "Fig.5h" and "Fig.5e" should be "Fig.5g" and "Fig.5h" ?

We thank the reviewer for careful reading of the text. We corrected these typos.

The sentence now reads: Then the lower edge state becomes unstable and emits 2 skyrmion-antiskyrmion pairs (Fig. 6g), which separate into two skyrmions that return to the lower edge and two skyrmions that move towards the upper edge (Fig. 6h).

REVIEWERS' COMMENTS:

Reviewer #1 (Remarks to the Author):

In the revised manuscript, the authors have addressed satisfactorily most of the comments raised by both the referees, and the manuscript has been improved significantly. I would like to recommend the present manuscript for publication.

Reviewer #2 (Remarks to the Author):

The referee checked that the referee's previous comments have been taken care of by the authors in some way or other. Hence, I now wish to recommend the publication of the manuscript in Nature Communications in the present form.